## Neglected Tropical Diseases

# High seroprevalence of antibodies to Dengue, Chikungunya, and Zika viruses in Dire Dawa, Ethiopia: A cross-sectional survey in 2024

**Daniel M. Parker**[1]*, **Werissaw Haileselassie**[2], **Temesgen Sisay Hailemariam**[3], **Arsema Workenh**[4], **Salle Workineh**[2], **Xiaoming Wang**[1], **Ming-Chieh Lee**[1], **Guiyun Yan**[1]

1 Joe C. Wen School of Population and Public Health; University of California, Irvine; Irvine, California, United States of America, 2 School of Public Health; College of Health Sciences; Addis Ababa University; Addis Ababa, Ethiopia, 3 Medical Diagnostic Laboratory, Tikur Anbessa Specialized Hospital; College of Health Sciences, Addis Ababa University; Addis Ababa, Ethiopia, 4 Dil Chora General Hospital, Dire Dawa Health Bureau, Dire Dawa, Ethiopia

* dparker1@hs.uci.edu

## Abstract

### Background

Aedes-borne diseases infect millions of people each year. In the last decade several arbovirus outbreaks have been reported in Ethiopia. Arbovirus diagnosis and surveillance is lacking, and the true burden is unknown.

### Methods

In this study we conducted a seroprevalence survey using a commercially available test kit that tests for immunological responses to IgM and IgG for dengue (DENV), Zika (ZIKV), and chikungunya (CHIKV) viruses in Dire Dawa city, eastern Ethiopia. A total of 339 individuals were sampled using a household-based clustered design. As a contextual comparison, a secondary survey was conducted among 180 individuals in Addis Ababa, where no Aedes-borne virus outbreaks have been reported.

### Findings

We found a high IgG seroprevalence for DENV (76%), CHIKV (44%), and ZIKV (38%), and <20% IgM seropositivity across all viruses. In contrast, minimal seropositivity was detected in Addis Ababa (where the highest seropositivity we found was to IgM for DENV at approximately 3%.) Age-specific trends showed early and widespread DENV exposure, with over half of the population seropositive by age 10. Quantitative antibody levels indicated strong correlation between DENV and ZIKV IgG, suggesting potential cross-reactivity. However, higher DENV IgG titers among ZIKV-positive individuals raise the possibility of true prior co-exposure. Intraclass correlation analyses revealed household-level clustering for DENV and CHIKV responses but not for ZIKV.

**Data availability statement:** The data for this study are available at: https://github.com/parker-group/DireDawa_Seroepi.

**Funding:** This study was partially funded through a UC Irvine Council on Research, Computing and Libraries (CORCL) pilot grant to DMP and through NIH grants D43 TW001505 and U19 AI129326 to GY. The funders had no role in study design, data collection and analysis, decision to publish, or preparation of the manuscript.

**Competing interests:** The authors have declared that no competing interests exist.

## Conclusions

These results suggest intense and possibly ongoing transmission of Aedes-borne viruses in Dire Dawa, particularly dengue and chikungunya. Apparent ZIKV exposure warrants cautious interpretation given the potential for cross-reactivity, but cannot be ruled out. Our findings underscore the need for improved arbovirus surveillance and diagnostic capacity in Ethiopia, especially in urban centers where competent vectors are established.

### Author summary

Mosquito-borne viruses such as dengue, chikungunya, and Zika pose growing public health challenges in many parts of Africa, but limited surveillance makes it difficult to assess their true burden. We conducted a serosurvey in Dire Dawa, a city in eastern Ethiopia that has experienced outbreaks of dengue and chikungunya. Blood samples from 339 individuals were tested for antibodies to dengue, chikungunya, and Zika viruses. We found high levels of prior exposure to dengue (76%) and chikungunya (44%), and observed apparent Zika virus antibodies in 38% of participants. Since antibodies can cross-react between related viruses, we interpret the Zika findings with caution and consider them hypothesis-generating. In contrast, we found low evidence of exposure in a separate sample from Addis Ababa, a city with no known history of arbovirus outbreaks. These results highlight the importance of community-based serosurveillance for detecting overlooked transmission and guiding targeted prevention strategies in urban African settings.

## Introduction

Aedes-borne diseases contribute significantly to the global burden of disease, and are especially prevalent in tropical and subtropical settings. *Ae. aegypti*, and to a lesser extent *Ae. albopictus*, are the major vectors of disease and three viruses of primary concern are dengue virus (DENV, from genus *Orthoflavivirus*), Zika virus (ZIKV, from genus *Orthoflavivirus*), chikungunya virus (CHIKV, from genus *Alphavirus*).

DENV, CHIKV, and ZIKV each represent significant public health challenges with distinct impacts and distributions. DENV, endemic in tropical and subtropical regions, is responsible for an estimated 400 million infections annually, with a subset of cases progressing to dengue hemorrhagic fever and dengue shock syndrome [1]. CHIKV was first detected in Tanzania in 1952 and in recent years has caused epidemics in Africa, Asia, Europe, and the Americas [2,3]. The disease is characterized by debilitating joint pain and arthritis that can last weeks to months, with significant economic implications on top of health impacts [4]. ZIKV was first detected in Uganda in 1947 and in the last decades there have been epidemics in Africa, Asia, the Americas, and the Pacific [5]. While most ZIKV infections are mild, the disease has been associated

with congenital abnormalities (e.g., microcephaly) when transmitted from mother to fetus [6–8]. Collectively, these mosquito-borne viruses cause substantial morbidity, impact quality of life, and strain healthcare systems globally.

While these viruses cause considerable disease globally, infections are presumptively diagnosed in much of the world, based on symptoms and without laboratory confirmation of the cause of disease. DENV and CHIKV infections result in similar symptoms and may be confused in some settings in the absence of laboratory confirmation. ZIKV infections are often asymptomatic. All three may be drastically under-diagnosed in some settings and surveillance is lacking in much of the African continent so that the true burden of these diseases is unknown.

Seroprevalence surveys offer the possibility to evaluate both the long- and short-term epidemiology of these diseases. IgM antibodies typically emerge shortly after infection (normally within days for each virus) and can persist for months. IgG antibodies typically emerge weeks after infection and can persist for years (in the case of dengue, may be lifelong). Dengue virus has four distinct serotypes (DENV-1 through DENV-4), and infection with one serotype typically provides lifelong immunity to that serotype but not to the others. Seroprevalence estimates from IgM are therefore normally indicative of recent epidemiology of these diseases whereas seroprevalence estimates of IgG are indicative of long-term dynamics. In some high transmission settings, most of the population will be seropositive before reaching adulthood [9,10]. Cross-reactivity, particularly between flaviviruses such as dengue and Zika, can complicate seroprevalence surveys that attempt to distinguish exposure to specific viruses [11].

Several dengue fever outbreaks have been documented in Ethiopia since 2013, including reported outbreaks in Dire Dawa in 2013 and 2015, and in other parts of East Ethiopia since [12–18]. Limited surveillance and diagnostic capacity mean that many cases likely go undetected or unconfirmed, making it difficult to precisely define the timing and extent of outbreaks. Chikungunya outbreaks have also been reported more recently in the Eastern part of the nation [19,20], and seropositivity among symptomatic individuals has been reported from the Northwest [21] and more recently in the Northeast part of the nation in patients with suspected malaria infections [22]. Dire Dawa, a major city in the Eastern part of Ethiopia has been repeatedly impacted by dengue infections and more recently (in 2019) by a major chikungunya outbreak that infected over 40,000 individuals [19]. Entomological surveys have confirmed the presence of *Ae. aegypti* in this city as well [23]. Research on Zika infections in this setting is scarce [24]. Surveillance is lacking for all three viruses.

The primary objective of this research was to conduct a pilot study assessing the seropositivity to DENV, CHIKV, and ZIKV in parts of Dire Dawa, Ethiopia that have a history of dengue outbreaks. Since age patterns in seropositivity can reflect both recent and historical transmission intensity, our goal was also to examine potential associations between age and seropositivity. We conducted secondary analyses of cross-reactivity.

## Data and methods

### Ethics statement

Ethics approval was obtained from the Institutional Review Board of the College of Health Sciences, Addis Ababa University. Verbal and written informed consent were obtained from all participants before participating in the study. Verbal and written assent/consent for participation for those under 18 years old was granted by their parent or legal guardian.

### Study location

Dire Dawa is a major trade city in Eastern Ethiopia. The city has a hot, arid climate with a hot season from February through June and two rainy seasons: a major one from July through September and a minor rainy season in March – May. Temperatures normally range from 14.4C to 32.7C. The wettest month is normally August, with a typical month experiencing a 12.7 cm of cumulative precipitation. The estimated population of the city in 2024 is 485,558 and the city is at approximately 1,276 meters elevation.

We conducted a seroprevalence survey in Dire Dawa in March 2024. A total of 339 individuals were recruited for the survey. At the time of study planning, there were no reliable seroprevalence estimates for arboviruses in Dire Dawa. We

therefore based our power analysis on an assumed 20% IgG seroprevalence, a 95% confidence level, and 5% absolute precision. Assuming a household-level clustered sampling design with an average household size of 4.5 individuals and an intra-class correlation coefficient (ICC) of 0.1, we estimated a design effect of 1.35. This yielded a target sample size of approximately 332 individuals. Although assuming 50% seroprevalence would have been more statistically conservative (as it maximizes the required sample size), such an approach was not feasible given our resource constraints. We specifically targeted kebeles (administrative units) where local public health officials reported previous dengue outbreaks based on their clinical and field experience. While not derived from formal outbreak reports or geolocated case data, this strategy was intended to increase the likelihood of detecting a serological signal. Households were selected using a household registry and a table of random numbers. In each selected household, all individuals were invited to participate regardless of age.

For comparison, we conducted a secondary seroprevalence survey on the outskirts of Addis Ababa in August 2024, using the same approach (selecting individuals by household). This secondary survey was not part of the primary research aim, but was instead intended as a contextual control to help interpret results from the main study in Dire Dawa. Addis Ababa is the capital and largest city in Ethiopia, with an estimated population in 2024 of 5,703,628. In contrast to Dire Dawa, Addis Ababa has a mild and temperate climate, rarely experiencing temperatures above 26.7C and typically ranging from 8.9C to 23.9C. The rainy season follows the same general pattern as Dire Dawa, but with August typically experiencing 20.3 cm of cumulative precipitation. Addis Ababa is at an elevation of approximately 2,355 meters.

We recruited 180 individuals for this secondary survey, targeting two kebeles in Addis Ababa. The survey was not intended to generate precise prevalence estimates, but rather to serve as a pragmatic reference point by applying the same diagnostic tests in a setting with no known history of Aedes-borne arbovirus outbreaks. Given the anticipated low exposure, we expected few if any seropositive results. Our goal was to contextualize test performance in a lower-risk area given our absence of laboratory-based confirmatory diagnostics.

## Seropositivity test

We used the Chembio DPP ZCD IgM/IgG system for the seroprevalence survey [25]. This system is an immunochromatographic test, meant for detection and differentiation of IgG and IgM antibodies to DENV, CHIKV, and ZIKV (i.e., each individual potentially testing positive for 6 different outcomes). Whole blood samples were obtained via finger prick using standard sterile lancets, with approximately 10 µL of capillary blood collected for each test. The assay was conducted at the point of care according to manufacturer instructions. After the sample and buffer were added to the test device, the Chembio DPP reader was used 10–15 minutes later to generate a digital readout of results.

The reader for the rapid diagnostic test also outputs a numeric value of the intensity of the immunological response. The numeric value corresponds to the intensity of the colored lines on the test device, which are proportional to the presence of antibodies. The reader uses photometry to measure light reflectance from the lines on the test strip, with the intensity of reflected light being inversely proportional to the relevant antibody responses. This quantitative measure provides an opportunity to look at patterns of cross-reactivity across viruses and immunological measures.

While this test has been evaluated in other geographic settings (e.g., Latin America [26], Southeast Asia [27]), no published validation studies have assessed its performance in African populations. As such, the sensitivity and specificity of this assay in our study setting are unknown, particularly in the presence of other flaviviruses.

## Questionnaire

The survey team visited each household with a tablet that was loaded with basic demographic questions (age and gender), test responses (both quantitative and binary), and test kits.

## Generalized estimating equations

We used a generalized estimating equations (GEE) model to estimate the overall seroprevalence to IgG and IgM for all three viruses, accounting for the household clustering design of the study by specifying an exchangeable correlation structure. We used bootstrapping to obtain robust confidence intervals around the seroprevalence estimates. We then used another GEE model to estimate age-specific seropositivity to IgG and IgM to each of the three viruses (DENV, CHIKV, and ZIKV). To model the relationship between age and seropositivity, we incorporated natural spline functions, allowing us to capture potential non-linear patterns in seropositivity across age. The model estimated the probability of being seropositive for each virus as a function of age, adjusted for household clustering. We then used these model estimates to generate and plot predicted probabilities of seropositivity across age groups. Confidence intervals for these predictions were calculated by deriving the standard errors of the linear predictor, which were then transformed to the probability scale using the logistic function. These predicted probabilities, along with their 95% confidence intervals, were plotted to visualize how IgG and IgM seropositivity changes with age for each virus, while accounting for clustering at the household level.

## Assessment of cross-reactivity

We used linear mixed models, with the raw quantitative immunological measures for each of the viruses and for both IgG and IgM as outcome variables, and with the other IgG or IgM measures for each virus as predictors, while accounting for household clustering using a random intercept for household. For models that used IgM measures as the outcome we also included IgG measures as covariates because we hypothesized that previous exposure to other viruses could potentialy impact immunological reactions to a more recent infection (e.g., previous exposure to DENV might influence immunological reaction to ZIKV).

While cross-reactivity between flaviviruses (DENV and ZIKV) is well-documented and informed our modeling approach, CHIKV (an alphavirus) was included as a comparison point, given its distinct viral lineage and expected lack of cross-reactivity. Published studies support this assumption, reporting minimal cross-reactivity between alphaviruses and flaviviruses [28]. This contrast helps reinforce the specificity of observed associations among the flaviviruses.

## Assessment of household clustering

The use of linear mixed models in our assessment of cross-reactivity also allowed us to investigate potential household clustering in immunological responses. To quantify household clustering we used the intraclass correlation coefficient (ICC). The ICC quantifies the proportion of total variance attributable to differences between households, relative to differences among individuals within the same household. The mixed models were fitted with random intercepts for households, allowing us to estimate the variance in responses across households. By comparing the random intercept variance to the total variance (random intercept variance plus residual variance), we calculated the ICC for each model. This provided insight into how much of the variation in IgG and IgM responses could be attributed to clustering within households versus individual-level variability.

# Results

## Overall seroprevalence

A total of 339 individuals were recruited for the seroprevalence survey in Dire Dawa (Table 1). Seropositivity to dengue virus IgG was 76% (95% Cis: 0.71 – 0.80). Chikungunya and Zika IgG seropositivity were 44% (0.38 – 0.50) and 38% (0.31 – 0.44), respectively. Seropositivity to IgM antibodies, typically indicating recent infections, was much lower (Table 2). Dengue and chikungunya IgM was approximately the same (0.17 (0.13 – 0.21) versus 0.18 (0.13 – 0.22), respectively). Zika IgM was much lower, at approximately 0.03 (0.02 – 0.05).

**Table 1. Crude serological results from Dire Dawa (March 2024) by age group, virus, and antibody type (IgG or IgM). Counts represent the number of individuals testing positive for IgG or IgM antibodies using the DPP ZCD rapid test. A total of 339 individuals were tested across all age groups.**

| Age Group | IgG | | | IgM | | | Total tested |
|---|---|---|---|---|---|---|---|
| | Dengue | Zika | Chikungunya | Dengue | Zika | Chikungunya | |
| 0-4 | 4 | 1 | 0 | 1 | 0 | 0 | 15 |
| 5-9 | 9 | 3 | 4 | 3 | 0 | 2 | 17 |
| 10-19 | 35 | 21 | 17 | 9 | 2 | 9 | 57 |
| 20-29 | 40 | 21 | 24 | 10 | 1 | 10 | 57 |
| 30-39 | 67 | 30 | 40 | 15 | 2 | 9 | 75 |
| 40-49 | 42 | 18 | 23 | 6 | 0 | 10 | 47 |
| 50-59 | 24 | 11 | 22 | 6 | 1 | 9 | 30 |
| 60-69 | 18 | 12 | 11 | 3 | 3 | 6 | 23 |
| 70+ | 14 | 9 | 8 | 2 | 2 | 1 | 18 |

**Table 2. Seropositivity estimates and confidence intervals, from GEE models using an exchangeable correlation structure to account for potential clustering in households. The data are from a seroprevalence survey done in Dire Dawa, Ethiopia in March 2024.**

| | Disease | Serporevalence Estimate | 95% Cis |
|---|---|---|---|
| IgG | Dengue | 0.76 | (0.71 − 0.80) |
| | Chikungunya | 0.44 | (0.38 − 0.50) |
| | Zika | 0.38 | (0.31 − 0.44) |
| IgM | Dengue | 0.17 | (0.13 − 0.21) |
| | Chikungunya | 0.18 | (0.13 − 0.22) |
| | Zika | 0.03 | (0.02 − 0.05) |

In comparison, few individuals from the secondary seroprevalence survey in Addis Ababa (August 2024) tested positive (not enough for accurate seroprevalence estimates). A total of 180 individuals were recruited in this survey (Table A in S1 Text) and out of them 5 tested positive for DENV IgG, 2 for ZIKV IgG, and 1 for CHIKV IgG. Six tested positive for DENV IgM, 1 tested positive for ZIKV IgM, and 1 tested positive for CHIKV IgM. Again, each of the ZIKV IgG positives were also DENV IgG positive. A total of 11 individuals were found seropositive (combining IgG and IgM, and for all three viruses); these individuals ranged in age from 25 to 36 (mean = 30.2; median = 28), and each reported travel to other regions in Ethiopia (five specifically mentioned travel to Dire Dawa).

## Age patterns in seropositivity

We first examined crude age group–specific seroprevalence estimates (Fig 1, Table 1), which showed a clear age-related increase in IgG seropositivity for all three arboviruses. These patterns were broadly consistent with modeled predictions using generalized estimating equations (GEE), which incorporated age as a continuous spline term and accounted for household clustering (Fig 2). By age 10, 52% of all sampled individuals were seropositive for IgG to DENV; this proportion rose to approximately 86% by age 40. For CHIKV, over half (51%) of individuals were seropositive by age 30, while ZIKV seropositivity reached 37% by age. The potential cross-over between IgM responses for dengue and chikungunya was likewise evident in these estimates (Fig 2), though the confidence bands around the estimates were sufficiently large to warrant caution in interpreting this result.

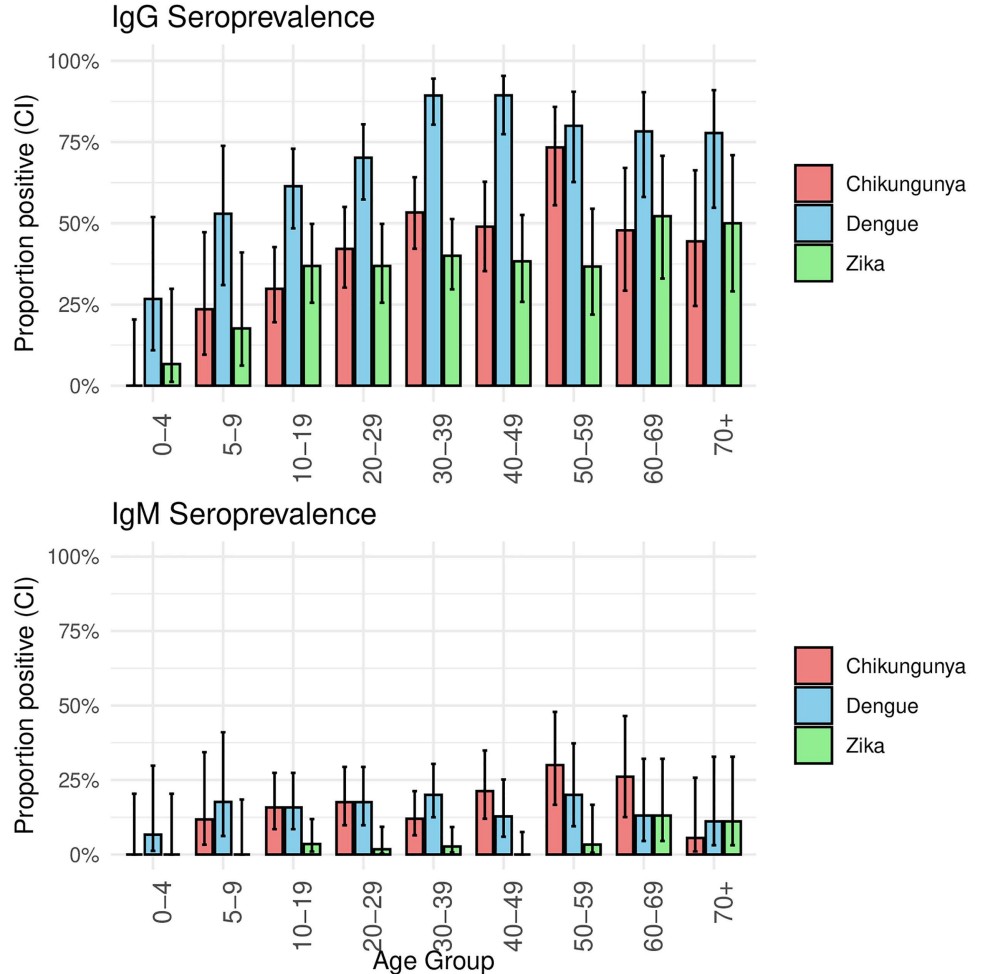

**Fig 1. Age-specific seroprevalence of IgG and IgM antibodies to dengue, Zika, and chikungunya viruses in Dire Dawa, Ethiopia (March 2024).** Bars represent the proportion of individuals in each age group who tested positive for each virus. Error bars indicate 95% confidence intervals calculated using the Wilson score method. IgG results reflect past exposure, while IgM results suggest more recent or acute exposure. A total of 339 individuals were included in the survey. Fig A in S1 Text presents age specific estimates with smaller age groups.

## Potential for cross reactivity

All Zika IgG positive individuals were also dengue IgG positive, suggesting a potential for cross-reactivity. Indeed, both are flaviviruses (i.e., genus *Orthoflavivirus*), and so cross-reactivity would not be surprising. A total of 253 of the tested individuals were dengue IgG seropositive and 126 were Zika IgG positive.

We used the raw quantitative values generated by the test reader as the outcome variables in our cross-reactivity analyses, rather than relying solely on the binary classifications of seropositivity. These values reflect the magnitude of the antibody signal detected by the commercial test kit, though they are not derived from laboratory-based titration methods such as PRNT (plaque-reduction neutralization test) or ELISA (enzyme-linked immunosorbent assay).

An analysis of the quantitative IgG responses for Zika and dengue showed that individuals who were categorized as Zika IgG seronegative but dengue IgG seropositive had very low Zika IgG values. Those who were seropositive for both Zika IgG and dengue IgG had much higher Zika IgG responses than individuals who were dengue IgG seropositive but

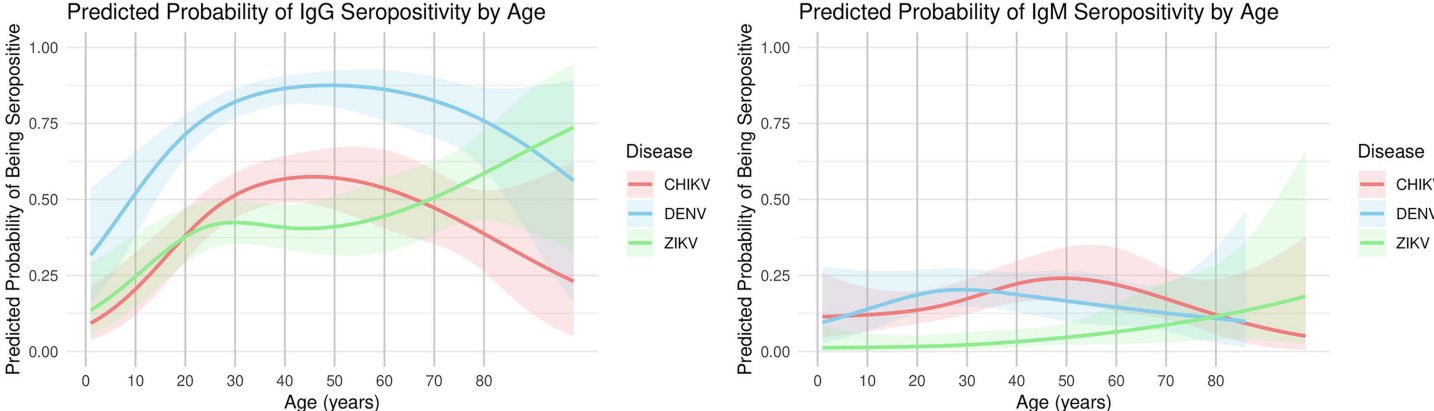

**Fig 2. Predicted probability of IgG seropositivity by age based on GEE models (Dire Dawa, March 2024).** Predicted probabilities of seropositivity for dengue, Zika, and chikungunya IgG are shown as smoothed curves generated from generalized estimating equations (GEEs), incorporating age as a natural spline (df = 3) and adjusting for household-level clustering. Shaded areas represent 95% confidence intervals. Estimates reflect modeled age-specific trends in seropositivity, rather than crude age group proportions.

Zika IgG seronegative (Fig B in S1 Text). Individuals who were dual-seropositive for dengue IgG and Zika IgG had significantly higher quantitative dengue IgG values compared to individuals who were dengue IgG seropositive but Zika IgG seronegative (Fig B in S1 Text). In general, cross-reactivity appeared stronger in the direction of ZIKV influencing DENV IgG levels than the reverse. We therefore hypothesize that Zika IgG seropositivity in this sample is not solely the result of cross-reactivity with dengue virus, but may reflect true prior infection with ZIKV. Furthermore, not all Zika IgM seropositive individuals were also dengue IgM seropositive, supporting this distinction. Given the lack of test validation in East African settings and the absence of confirmatory assays such as PRNT, these patterns should be interpreted cautiously. The observed ZIKV seropositivity may reflect true prior exposure or cross-reactive antibody responses, particularly among individuals with high DENV IgG levels.

We generated a Spearman's correlation matrix from all possible immunological outcomes (Fig 3). The strongest associations were between Zika IgG and dengue IgG, chikungunya IgG and chikungunya IgM, followed by Zika IgG and dengue IgM, and dengue IgG and IgM.

Our linear mixed models analyses (Tables B and C in S1 Text) showed that Zika IgG was strongly associated with dengue IgG values (Table B in S1 Text). Chikungunya IgG was also associated but the association was smaller than with Zika IgG. Dengue IgG was strongly associated with Zika IgG. Chikungunya IgG was negatively associated with Zika IgG. Finally, Zika IgG was positively associated with chikungunya IgG values while dengue IgG was negatively associated with chikungunya IgG.

Zika IgM was positively associated with dengue IgM, as was chikungunya IgM (Table C in S1 Text). Likewise, dengue and chikungunya IgM values were positively associated with Zika IgM. Finally, dengue and Zika IgM were both positively associated with chikungunya IgM. IgG values were statistically associated with each IgM outcome, but their effect sizes were small.

## Household clustering

The ICCs (intraclass correlation coefficients) indicated varying degrees of household clustering for the different viruses and immune responses. For DENV, we observed moderate clustering in IgG responses (ICC = 10.3%) and low clustering for IgM responses (ICC = 1.8%), suggesting some shared exposure or immunity patterns within households. CHIKV showed similar IgG clustering (ICC = 7.1%) and a stronger clustering effect in IgM responses (ICC = 14.0%). This may

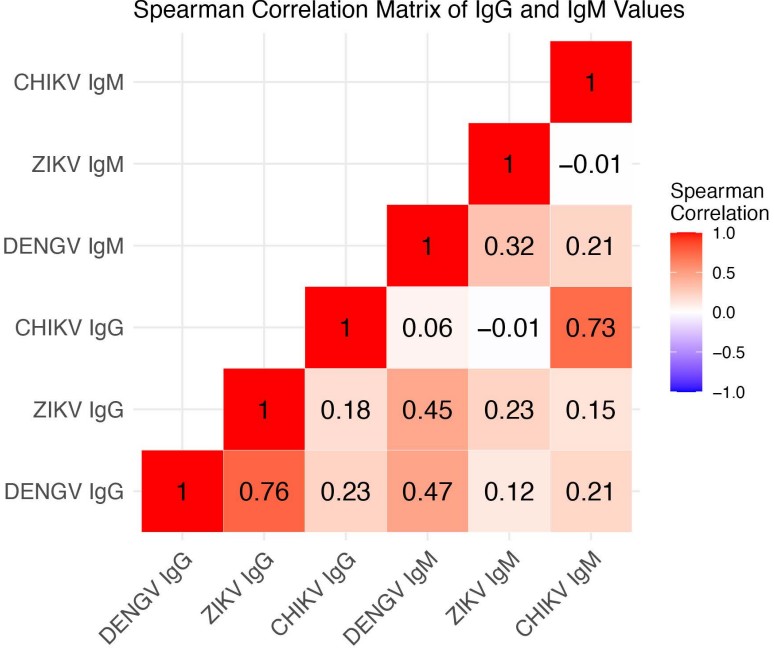

**Fig 3. Spearman correlation matrix of quantitative IgG and IgM levels for each arbovirus.**

indicate recent transmission events that clustered in space and time. ZIKV responses showed no detectable household clustering for either IgG or IgM (ICC = 0 for both). This lack of clustering may reflect true absence of household-level transmission - consistent with a lower force of infection - or could be the result of low statistical power, given the low frequency of ZIKV seropositivity in our sample. If ZIKV transmission were ongoing, some degree of clustering might be expected; the lack of such clustering could suggest minimal circulation of the virus or that the observed IgG responses are driven by cross-reactive antibodies unrelated to recent, spatially structured exposure.

## Discussion

Using a cross-sectional seroprevalence survey from Dire Dawa, Ethiopia, we found and here report a high seroprevalence to three arboviruses of concern: DENV, CHIKV, and ZIKV. These results existed for both IgG and IgM suggesting both a history of widespread exposure to these viruses, and recent exposure. The age patterns for DENV and CHIKV IgG seroprevalence in particular suggest that most of the population in this setting has been exposed to these viruses by age 30 (over 50% seroprevalence to CHIKV IgG and over 80% to DENV IgG). We also found evidence of household clustering with regard to DENV and CHIKV exposure, but not ZIKV exposure.

While approximately 25% of all children ages 0 – 4 were seropositive for IgG to DENV, none were seropositive for IgG to CHIKV. In 2019 there was a major CHIKV outbreak in this setting, with over 40,000 individuals diagnosed with the disease [19]. It is therefore possible that much of the seroprevalence to CHIKV IgG that we found was the result of this previous epidemic, or to the 2019 epidemic and others. We did, however, find individuals in most ages (but not the youngest age group) who were seropositive for IgM to CHIKV, suggesting that there has potentially been some recent transmission of this disease or a related alphavirus, among individuals from the study location. IgM to Chikungunya has been shown to persist in some individuals over long periods of time (28 months post infection [29]) but we believe this is unlikely to explain the IgM seropositivity to CHIKV from our survey (now 5 years after the major 2019 epidemic).

While dengue and chikungunya epidemics have previously been reported in Ethiopia, exposure to Zika is poorly characterized because of limited surveillance. A large, nationally representative serosurvey conducted using multistage sampling and confirmatory plaque reduction neutralization tests (PRNT) detected low levels of flavivirus antibodies, including 0.4% ZIKV IgG positivity [24]. A smaller, community-level survey in Gambella (in West Ethiopia) estimated seroprevalence of 2.9, 15.6, and 27.3% seroprevalence for yellow fever, CHIKV, and ZIKV, respectively [30], though the absence of confirmatory testing raises concerns about potential cross-reactivity. Elsewhere in Ethiopia, most serological studies have focused on hospital-based or febrile populations, which may not accurately reflect broader community-level transmission.

In contrast, our survey employed a population-based household sampling strategy in a setting with known dengue and chikungunya outbreaks. We found high IgG seroprevalence, particularly for dengue, suggesting intense and sustained local transmission in Dire Dawa. These estimates may even underestimate true exposure given prior evaluations indicating low sensitivity of the rapid test we used [27]. Our secondary survey in Addis Ababa, where no outbreaks have been reported, served as a pragmatic control. The near-absence of seropositivity in this study population further supports the interpretation that the high values observed in Dire Dawa reflect localized transmission rather than test artifact.

We also observed substantial IgG reactivity to ZIKV in Dire Dawa. While this may indicate past ZIKV exposure, interpretation is complicated by well-documented cross-reactivity among flaviviruses, particularly between ZIKV and DENV. Without confirmatory tests we cannot definitively attribute these results to ZIKV infections. Yellow fever virus, another flavivirus, is endemic to parts of Ethiopia, and yellow fever vaccination can induce cross-reactive antibodies. However, there have been no yellow fever vaccination campaigns in Dire Dawa or Addis Ababa, making it unlikely to explain the observed ZIKV reactivity in our sample. Notably, individuals who were ZIKV IgG–positive had higher DENV IgG titers compared to those who were ZIKV IgG–negative. This could reflect true sequential infections or cross-reactive antibody responses; it may also support the hypothesis that prior ZIKV exposure influences DENV IgG responses. However, an equally plausible explanation is that high DENV IgG itself increases the probability of a false-positive ZIKV result. We therefore interpret these ZIKV IgG findings cautiously and view them as hypothesis-generating, underscoring the need for confirmatory investigation of arbovirus exposure and co-exposure in this setting.

One limitation of our work is the use of a single screening kit. It might have been preferable to have another test (e.g., ELISA) for comparison on at least a subsample. A few other studies have used the same test kit in research settings. One study suggested general low sensitivity of this test in comparison to ELISA. However, specificity was high (sensitivity/specificity: DENV 43.92/100%; ZIKV 25.86/98.81%; CHIKV 37.78/99.35% [27]). These evaluations, however, were not conducted in East African populations and no published validation studies exist for this test in East African settings. If the same pattern holds true in this setting it would indicate that the estimated seroprevalence is an underestimate [27]. Another study focusing on IgM alone found sensitivity/specificity of the ZCD test to ZIKV IgM was 79.0/97.1%; DENV was 90.0/89.2%; and for CHIKV it was 90.6/97.2% [26]. We take the results of these previous studies to suggest that for ZIKV and CHIKV in particular, our results may be underestimates. However, the absence of gold-standard confirmatory tests in our setting limits our ability to assess test performance directly. This limitation is particularly important when interpreting ZIKV results, which may be influenced by cross-reactivity with other flaviviruses (including DENV).

Cross reactivity may have influenced some of our results. DENV and ZIKV are both flaviviruses, and cross-reactivity between the two is well-known but not completely understood [11,31]. If other alphaviruses are circulating in this setting, they could potentially lead to false-positive CHIKV seropositivity. Although we observed patterns consistent with differential responses (e.g., higher DENV responses among ZIKV positives), these may still reflect cross-reactive antibody responses rather than true prior ZIKV infection.

These complexities have direct implications for surveillance and clinical interpretation. In the absence of molecular confirmation or PRNT-based serotyping, distinguishing true infections from cross-reactive responses is difficult. Misclassification could compromise seroprevalence estimates and introduce uncertainty into outbreak detection, risk mapping, and

vaccine eligibility assessments. This highlights the importance of continued assay validation, ideally in local populations, and careful interpretation of serological survey data in flavivirus-endemic settings.

Cross-reactivity can have clinical implications, as is already well-known within DENV infections across serotypes and has been hypothesized between DENV and ZIKV infections. While DENV infections are known to provide long-term immunity, the immunity is serotype specific and infection by another serotype can lead to more severe disease outcomes (including death) through antibody dependent enhancement (ADE) [32–34]. ADE may also occur across flaviviruses – with the potential for previous ZIKV infection leading to severe dengue disease even in the primary infection by DENV [35]. Current dengue vaccines are only provided to populations that already have previous exposure to DENV (i.e., the vaccines should only be provided to seropositive individuals) as vaccinating individuals who are seronegative can increase their risk of severe disease when they are infected by DENV [36].

Cross-reactivity is further complicated by the fact that these three viruses are all vectored by the same mosquito (e.g., *Ae. aegypti*), meaning that individuals who have been infected with dengue in this setting are likely to also have been at risk of chikungunya (which has been confirmed in this setting) and Zika virus infection, if those viruses were present or introduced. Although *Ae. aegypti* has been documented in this region, data on vector abundance, seasonality, and human-vector contact patterns remain scarce. This pilot study was designed in part to address that gap and to generate preliminary evidence that can motivate more detailed entomological and behavioral investigations in future work. That is, individuals in this setting who acquire dengue infections are likely to also be at risk of CHIKV or ZIKV infections, if those viruses are present, given the shared mosquito vector that is present in this setting.

This seroprevalence survey was designed to evaluate age-specific seroprevalence in kebeles in Dire Dawa known to have previously had dengue outbreaks. While we used a clustered random sampling approach (randomly selecting households within the kebeles), our results may not be representative of the entire city of Dire Dawa. We also do not have precise information on the total population size or geographic areas represented by the selected kebeles, which limits our ability to generalize these findings to the broader urban population. It would be valuable to conduct further surveys that would be representative of the city, as well as in other parts of Ethiopia.

The observed age-specific IgG seroprevalence patterns offer important epidemiological insight. The rapid rise in DENV IgG positivity by age 10 (Figs 1 and 2) likely reflects active or recent transmission, as seropositivity among young children must result from infections within their relatively short lifespans. In contrast, slower age-related increases in CHIKV or ZIKV IgG suggest a lower force of infection or more recent introduction of these viruses. The smooth, progressive increase in DENV IgG with age also supports the interpretation that exposure is accumulating gradually, consistent with endemic transmission rather than isolated outbreaks. While maternally derived antibodies may explain some reactivity in very young children, the broader pattern suggests widespread exposure during early childhood. Although the ZIKV findings are also intriguing, we remain cautious in interpreting those results because of the potential for cross-reactive antibody responses with DENV.

We also found evidence of household level clustering (from the ICCs) for DENV and CHIKV IgG responses and for CHIKV IgM responses. In contrast, ZIKV responses showed no measurable household clustering. These patterns could reflect differing transmission dynamics or the timing of exposure across viruses. Household clustering in IgG or IgM responses may point to shared environmental exposures such as mosquito breeding sites near houses, or to household-level behaviors and socioeconomic factors that influence risk of infection. Future interventions or surveillance efforts might benefit from recognizing and targeting such within-household transmission patterns.

In summary, we found evidence of widespread exposure to DENV, CHIKV, and ZIKV in Dire Dawa, Ethiopia. This aligns with reported outbreaks of dengue fever in this setting, and of the massive chikungunya fever outbreak in 2019. We also found evidence of exposure to ZIKV, which has rarely been reported from this nation. Not only did we find evidence of previous exposure to ZIKV, we also found evidence of recent exposure (i.e., IgM seropositivity) to all three arboviruses. While we cannot rule out the potential for cross-reactivity influencing our results, they are suggestive of a large risk of infection

with *Ae. aegypti*-transmitted viruses in a setting where this important vector of disease is known to be present [23]. Repeated infections can in some cases lead to more severe disease outcomes, for example through ADE. More research into arboviral diseases and their mosquito vectors in this setting is warranted both to confirm our results and to develop a better understanding of the distribution of different arboviral infections, any potential age and gender associated risk factors, and all so that public health interventions can be formulated to address these diseases. As the world has experienced pandemic levels of Aedes-borne disease several times since 2019, it is increasingly important to have increased surveillance for these diseases and for public health agencies to plan accordingly.

## Supporting information

**S1 Text.** **Fig A.** *Age-specific seroprevalence by arbovirus and for IgG and IgM Page 2*. **Fig B.** IgG responses for ZIKV and DENV by test status Page 3. **Table A.** Crude serological results from Addis Ababa by age group, virus, and antibody type Page 4. **Table B.** Linear mixed models for IgG (random intercept for household) Page 5. **Table C.** Linear mixed models for IgM (random intercept for household) Page 6. **Description A. Power Analysis.** Sample size calculation methodology for Dire Dawa sero-survey Page 7. **Description B. Statistical Modeling of Age-Seropositivity Relationships.** GEEs with splines for IgG and IgM Page 8. **Description C. Linear Mixed-Effects Modeling of Antibody Responses and Cross-Reactivity.** Page 9.
(DOCX)

## Acknowledgments

We are thankful for the community members and community health workers in Dire Dawa and Addis Ababa who participated in this research.

## Author contributions

**Conceptualization:** Daniel M Parker, Guiyun Yan.

**Data curation:** Werissaw Haileselassie, Temesgen Sisay Hailemariam, Arsema Workenh, Salle Workineh.

**Formal analysis:** Daniel M Parker.

**Funding acquisition:** Daniel M Parker, Guiyun Yan.

**Investigation:** Werissaw Haileselassie, Temesgen Sisay Hailemariam, Arsema Workenh, Salle Workineh.

**Methodology:** Daniel M Parker, Guiyun Yan.

**Project administration:** Werissaw Haileselassie.

**Supervision:** Daniel M Parker, Guiyun Yan.

**Validation:** Daniel M Parker, Guiyun Yan.

**Visualization:** Daniel M Parker, Mingh-Chieh Lee.

**Writing – original draft:** Daniel M Parker.

**Writing – review & editing:** Daniel M Parker, Werissaw Haileselassie, Temesgen Sisay Hailemariam, Arsema Workenh, Salle Workineh, Xiaoming Wang, Mingh-Chieh Lee, Guiyun Yan.

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
