## [Decision Letter · Decision Letter 0]

High Seroprevalence to Aedes -borne arboviruses in Ethiopia: a Cross-sectional Survey in 2024

Dear Dr. Parker,

Thank you for submitting your manuscript to PLOS Neglected Tropical Diseases. After careful consideration, we feel that it has merit but does not fully meet PLOS Neglected Tropical Diseases's publication criteria as it currently stands. Therefore, we invite you to submit a revised version of the manuscript that addresses the points raised during the review process.

Please submit your revised manuscript within 60 days Jun 10 2025 11:59PM. If you will need more time than this to complete your revisions, please reply to this message or contact the journal office at plosntds@plos.org. Please include the following items when submitting your revised manuscript:

We look forward to receiving your revised manuscript.

Kind regards,

Expedito J A Luna, MD

Academic Editor

Michael Holbrook

Section Editor

Shaden Kamhawi

co-Editor-in-Chief

Paul Brindley

co-Editor-in-Chief

**Additional Editor Comments (if provided):**

Dear Dr. Parker,

Your manuscript has been carefully reviewed, and major revision are required for acceptance. Please consider the issues raised by the reviewers.

**Journal Requirements:**

At this stage, the following Authors/Authors require contributions: Daniel M Parker, Werissaw Haileselassie, Temesgen Hailemariam, Arsema Workenh, Salle Workineh, Xiaoming M Wang, Mingh-Chieh M Lee, and Guiyun M Yan. Please ensure that the full contributions of each author are acknowledged in the "Add/Edit/Remove Authors" section of our submission form.

4) We do not publish any copyright or trademark symbols that usually accompany proprietary names, eg ©,  ®, or TM  (e.g. next to drug or reagent names). Therefore please remove all instances of trademark/copyright symbols throughout the text, including:

- ® on pages: 6, 18, and 19.

5) Please upload all main figures as separate Figure files in .tif or .eps format. For more information about how to convert and format your figure files please see our guidelines: 

6) We have noticed that you have uploaded Supporting Information files, but you have not included a list of legends. Please add a full list of legends for your Supporting Information files after the references list.

7) When completing the data availability statement of the submission form, you indicated that you will make your data available on acceptance. We strongly recommend all authors decide on a data sharing plan before acceptance, as the process can be lengthy and hold up publication timelines. Please note that, though access restrictions are acceptable now, your entire data will need to be made freely accessible if your manuscript is accepted for publication. This policy applies to all data except where public deposition would breach compliance with the protocol approved by your research ethics board. If you are unable to adhere to our open data policy, please kindly revise your statement to explain your reasoning and we will seek the editor's input on an exemption. Please be assured that, once you have provided your new statement, the assessment of your exemption will not hold up the peer review process.

8) Please amend your detailed Financial Disclosure statement. This is published with the article. It must therefore be completed in full sentences and contain the exact wording you wish to be published.

1) State what role the funders took in the study. If the funders had no role in your study, please state: "The funders had no role in study design, data collection and analysis, decision to publish, or preparation of the manuscript.".

**Reviewers' Comments:**

Reviewer's Responses to Questions

**Key Review Criteria Required for Acceptance?**

**Methods**

-Are the objectives of the study clearly articulated with a clear testable hypothesis stated?

-Is the study design appropriate to address the stated objectives?

-Is the population clearly described and appropriate for the hypothesis being tested?

-Is the sample size sufficient to ensure adequate power to address the hypothesis being tested?

-Were correct statistical analysis used to support conclusions?

-Are there concerns about ethical or regulatory requirements being met?

Reviewer #1: (No Response)

Reviewer #2: Would be beneficial to include the model equations and power calculation? Why were the number of individuals tested?

Why were only districts with historical dengue targeted. If it was for resources thats fine but justify.

Should consider incorporating published sensitivity and specificity data for the serologic test into the data analysis to account for it.

It does not make sense biologically to test for IgG predicting IgM (Line 238). Or justify it.

Need a table showing how many people in each age were tested.

Reviewer #3: The objectives of the study are clearly stated. The population is clearly described and appropriate.

It is not completely clear what was the purpose of having a second serosurvey site. You present the general seroprevalence in Addis Ababa but did you use this information in any other way? How was helpful? How was used?

No concerns relate to ethical or regulatory requirements.

**Results**

-Does the analysis presented match the analysis plan?

-Are the results clearly and completely presented?

-Are the figures (Tables, Images) of sufficient quality for clarity?

Reviewer #1: (No Response)

Reviewer #2: Need to present actual results of serological testing in table form not just estimates.

Reviewer #3: Yes, the results match the analysis plan. The results can be improved by presenting the crude numbers obtained for each serosurvey site and for clarifying is you are including just one site or both in each of your analysis and tables and graphs.

**Conclusions**

-Are the conclusions supported by the data presented?

-Are the limitations of analysis clearly described?

-Do the authors discuss how these data can be helpful to advance our understanding of the topic under study?

-Is public health relevance addressed?

Reviewer #1: (No Response)

Reviewer #2: Need to better address/discuss the limitation of the sensitivity and specificity of the Chembio test to get at cross reactivity.

Need to further elaborate on the mosquito biology and human behavior driving transmission in these areas to explain the results.

Need to discuss the quantitative break-points/cut-offs for what a positive is. (Lines 214-218) Are these derived from PRNT assays, ELISA, how? Even though this is not the point of your study you are using the quantitative values in your analysis and thus need to introduce them.

Lines 240 - 247: need to better articulate/explain this in the conclusions. If you have household clustering of denv why not zikv? You would likely see it if there was enough ZIKV transmission or if sample size was larger.

Line 262: syntax- you are not "seropositive to CHIKV IgM" you are seropositive for IgM to CHIKV.

Line 269: citation 21 needs context/elaboration. This whole paragraph needs to be reworked to have direction.

Line 313: Seroprevelance does not indicate burden of disease when the viral infections are largly asymptomatic and do not produce disease. It would be better to discuss this as risk of infection. If you want to discuss potential burden without surveying symptomatic patients then incorporate published data on the rate of disease resulting from infection.

Reviewer #3: More discussion about how the known performance of the used test (especially specificity) helps supporting your results and your claim that Zika results are not an artifact of cross-reactivity with dengue.

**Editorial and Data Presentation Modifications?**

Reviewer #1: (No Response)

Reviewer #2: Need to present actual testing results by age and area not just estimates in table form. Put the results in tables.

Some minor syntax and grammatical errors.

Stay consistent with metric units throughout.

A diagram explaining your cross reactivity hypothesis is needed.

Reviewer #3: As Yellow Fever is also an arbovirus transmitted by Ae. aegypti of high interest in the African continent, and to make your paper easier to find for those interested in DENV, CHIKV and ZIKV, consider using the specific names of the viruses instead of “Aedes-borne arboviruses” in your title.

Line 52: suggest including the word “virus” for each of them. “are” is repeated.

Line 55: you don’t need the word “viruses” here

Line 58: The current WHO clinical/severity classification is dengue, dengue without warning signs, and severe dengue. If you are including DHF and DSS because of country specific guidelines, I suggest you clarify that is the reason.

Line 69: Zika infections

Line 73: Suggest IgM antibodies instead of ‘antibodies to IgM’

Line 75: Consider including specifics about the 4 dengue virus types here

Line 78: Cross-reactivity is a concern for dengue and Zika viruses, but not for chikungunya

Line 80: consider adding additional details regarding the specific years for these outbreaks, this is specially important since you are doing age-related analyses

Line 101: you include the total number of participants recruited in the results

Line 104: It would be helpful to expand on this, did you have access to geolocation data of previous cases? City level official reports? Something else?

Line 106: Does “secondary” means anything specific here or refers to a second serosurvey? When was this survey conducted?

Line 112: Do you have information on the approximated area and population you covered for each of the serosurveys? I am assuming you did not select the full area of either city as they are of a considerable size.

Line 114: I don’t think you are measuring exposure to Aedes aegypti with this study, I suggest modifying this sentence.

Line 116: But how was this helpful or used for your analysis?

Line 119: Is this a fingertip prick test? Include this information in the methods.

Line 180: Suggest reporting your results in a consistent manner, either with % or with proportions throughout the document.

Line 202: Suggest reporting these results in increasing age groups (age 10, age 20, etc)

Line 210: Is this for all participants in both serosurveys? This is not clear for the Dire Dawa results section starting in line 180.

Line 124: this paragraph is confusing. Using negative and positive in a consistent manner (instead of “non positive”) might help with clarity.

Line 153: Chikungunya is an alphavirus and should not have any cross-reactivity concerns with the other two flaviviruses included in this analysis.

Line 217: Consider moving supplemental figure 2 to the main manuscript as this would help readers interpret your results in a much faster and easier manner.

Line 220: What does “reactive” means? Positive?

Line 257: Did you have infants under 6 months? What were their results for IgG? Thinking about maternal IgG transfer

Line 269: Check this sentence, seems to be incomplete

Line 270: Not sure this information is relevant. Overall, this whole paragraph can be reorganized and reworded.

Tables and Figures: Make sure that titles are complete, including place and time.

Figure 1: Since you are reporting the probability by age using 10, 20, 30, etc, consider having those ages as your x axis markers or adding something in the lines that guides to reader to the predicted probability of those ages.

**Summary and General Comments**

Reviewer #1: Major commetns:

The introduction does not adequately contextualize arboviral epidemiology in Ethiopia, specifically for Zika virus. More detail on existing studies or surveillance gaps is needed.

While the manuscript explains the selection of Dire Dawa due to its history of arbovirus outbreaks, it does not sufficiently justify the choice of Addis Ababa as the comparison site. Why was Addis Ababa selected despite its lack of reported outbreaks? Provide a clearer rationale for its inclusion and discuss how this might affect the study's conclusions.

The manuscript briefly discusses household-level clustering effects but does not explore the implications of this finding. For example, could household clustering suggest shared environmental risk factors, such as mosquito breeding sites or socioeconomic status? Discuss how these findings could guide targeted interventions.

Although the study acknowledges potential cross-reactivity between DENV and ZIKV IgG results, it does not explore how this might affect public health recommendations. Could these findings lead to diagnostic challenges or misclassification in clinical and epidemiological settings? Consider elaborating on strategies to address such issues.

The age-specific seroprevalence results are intriguing, particularly the rapid increase in IgG seropositivity for DENV by age 10. However, the manuscript does not delve into why children might exhibit such high seroprevalence early in life. Discuss potential exposure routes, such as maternal antibody transfer, vector density, or environmental factors.

The manuscript refers to seroprevalence studies from other regions but does not perform a detailed comparison. How do the findings from Dire Dawa and Addis Ababa align or differ from seroprevalence rates reported in other African countries or similar ecological settings?

Minor comments:

Many mistakes in the manuscript. Here is a part of it.

Line 31: “that that”

Line 52: “are are”

Line 55: “DENV, CHIKV, and ZIKV viruses”

Line 68: “Dengue and chikungunya infections”

Line 242: “DENGV”

Line 244: “Zika”

Line 255: “DENGV”

Line 362: lack “Figure 2”

Reviewer #2: Study has potential for meaningful publication highlighting the need for improved arboviral surveillance on the African continent but is missing critical discussion on the limitations of the serological test used in the study, data presentation, and articulation of what biologically is driving the results. Also, justification of various methodologies is needed.

Reviewer #3: This manuscript by Parker et al describe the results of dengue, chikungunya and Zika seroprevalence surveys conducted in Ethiopia. The information included in the manuscript is of interest for the field as very limite information on these arboviral diseases transmission is available for the region and the country. Overall, the manuscript is well written, althought it can benefit from some clarifications in the methods and results sections, and additional discussion details. I have included some specific comments for the authors.

PLOS authors have the option to publish the peer review history of their article (what does this mean? ). If published, this will include your full peer review and any attached files.

**Do you want your identity to be public for this peer review?** For information about this choice, including consent withdrawal, please see our Privacy Policy .

Reviewer #1: No

Reviewer #2: No

Reviewer #3: No

**Figure resubmission:**

**Reproducibility:**



---

## [Editor Report · Decision Letter 1]

Dear Prof. Parker,

We are pleased to inform you that your manuscript 'High Seroprevalence of Antibodies to Dengue, Chikungunya, and Zika Viruses in Dire Dawa, Ethiopia: A Cross-Sectional Survey in 2024' has been provisionally accepted for publication in PLOS Neglected Tropical Diseases.

Best regards,

Expedito J A Luna, MD

Academic Editor

Michael Holbrook

Section Editor

Shaden Kamhawi

co-Editor-in-Chief

Paul Brindley

co-Editor-in-Chief

You have properly addressed the issues raised by the reviewers. I recommend acceptance of the manuscript.
